# A Bioassay-Guided Fractionation of Rosemary Leaf Extract Identifies Carnosol as a Major Hypertrophy Inducer in Human Skeletal Muscle Cells

**DOI:** 10.3390/nu13124190

**Published:** 2021-11-23

**Authors:** Sylvie Morel, Gérald Hugon, Manon Vitou, Marie Védère, Françoise Fons, Sylvie Rapior, Nathalie Saint, Gilles Carnac

**Affiliations:** 1Laboratoire de Botanique, Phytochimie et Mycologie, Faculté de Pharmacie, CEFE, Univ Montpellier, CNRS, EPHE, IRD, 34090 Montpellier, France; sylvie.morel@umontpellier.fr (S.M.); manon.vitou@umontpellier.fr (M.V.); francoise.fons@umontpellier.fr (F.F.); sylvie.rapior@umontpellier.fr (S.R.); 2PhyMedExp, Univ Montpellier, CNRS, INSERM, 34090 Montpellier, France; gerald.hugon@inserm.fr (G.H.); marie.ved65@gmail.com (M.V.); nathalie.saint@inserm.fr (N.S.)

**Keywords:** *Salvia rosmarinus*, stem cells, diterpene, proteasome, muscle mass

## Abstract

A good quality of life requires maintaining adequate skeletal muscle mass and strength, but therapeutic agents are lacking for this. We developed a bioassay-guided fractionation approach to identify molecules with hypertrophy-promoting effect in human skeletal muscle cells. We found that extracts from rosemary leaves induce muscle cell hypertrophy. By bioassay-guided purification we identified the phenolic diterpene carnosol as the compound responsible for the hypertrophy-promoting activity of rosemary leaf extracts. We then evaluated the impact of carnosol on the different signaling pathways involved in the control of muscle cell size. We found that activation of the NRF2 signaling pathway by carnosol is not sufficient to mediate its hypertrophy-promoting effect. Moreover, carnosol inhibits the expression of the ubiquitin ligase E3 Muscle RING Finger protein-1 that plays an important role in muscle remodeling, but has no effect on the protein synthesis pathway controlled by the protein kinase B/mechanistic target of rapamycin pathway. By measuring the chymotrypsin-like activity of the proteasome, we found that proteasome activity was significantly decreased by carnosol and Muscle RING Finger 1 inactivation. These results strongly suggest that carnosol can induce skeletal muscle hypertrophy by repressing the ubiquitin-proteasome system-dependent protein degradation pathway through inhibition of the E3 ubiquitin ligase Muscle RING Finger protein-1.

## 1. Introduction

In humans, skeletal muscle represents ~40% of the body mass. Due to its ability to generate strength, muscle is required for mobility, breathing, and posture. Skeletal muscle also plays a key role as a metabolism regulator through the use of a large quantity of glucose and lipids, particularly during exercise. Many pathological (e.g., myopathies, cancer, diabetes) and non-pathological (e.g., sedentary lifestyle, bed rest, immobilization, spaceflights) conditions lead to loss of muscle mass and consequently to reduction of the functional capacities and life expectancy [1]. Skeletal muscle atrophy may also result in sarcopenia (i.e., the loss of muscle mass, size, strength and functionality because of aging). The importance of proper muscle function in human health is supported by data demonstrating that skeletal muscle strength is inversely associated with all-cause mortality in men [2]. Therefore, reduction of muscle loss may contribute to improve the quality of life and to attenuate chronic diseases and mortality. Currently, exercise and nutritional approaches based on supplementation of essential amino acids or their metabolites (e.g., β-hydroxy-β- methyl butyrate, citrulline, ornithine, and vitamin D) seem the most effective preventive measures to limit the muscle mass loss associated with diseases and aging. However, compliance with exercise programs and their implementation are difficult. These are partly the consequence of patient pathologies (fatigue, pain) which limits their participation in long-term physical activity compared to people without chronic illness [3,4]. Pharmaceutical groups have recently taken an interest in the loss of muscle mass and have developed their drug candidates, such as anti-myostatin antibodies or myostatin inhibitors and a selective modulator of the androgen receptor. However, some studies already showed that these molecules have side effects and are not very effective in humans [5,6,7].

Different signaling pathways control the balance between protein synthesis and degradation in skeletal muscle. Two main pathways coordinate the renewal of contractile proteins: the phosphoinositide 3-kinase/protein kinase B/mammalian target of rapamycin complex (PI3K/Akt/mTOR) pathway (the major signaling cascade of muscle hypertrophy), and the FOXO family pathway (transcription factors that control the expression of atrogenic and autophagy genes involved in proteasome-mediated degradation and autophagy, respectively). Activation of the PI3K/Akt pathway stimulates protein synthesis via its anabolic targets (mTOR, the protein S6 kinase 1 and the binding protein eIF4E) and blocks the FOXO proteolysis pathways mediated by E3 ubiquitin ligases of the muscle RING finger protein-1 (MuRF1, TRIM63) and muscle atrophy F-box (MAFbx) families, and consequently the proteasome activity [1,7,8]. Recent studies have shown that natural molecules found in plants or fruits could be used for the treatment or prevention of skeletal muscle atrophy by stimulating protein synthesis and/or inhibiting protein degradation [9,10,11,12,13,14].

Plants represent an important source of bioactive molecules and are at the origin of two-thirds of the currently used drugs [15]. Rosemary (*Salvia rosmarinus* Schleid., synonym: *Rosmarinus officinalis* L., Lamiaceae) stem and leaf extracts contain many different phenolic compounds, including flavonoids and phenolic diterpenes and triterpenes (Borras-Linares et al., 2014), with many major biological properties (antidiabetic, anti-inflammatory, antioxidant, and anticancer) [16,17,18,19,20]. The antioxidant activities of rosemary leaf extracts are mainly attributed to the phenolic diterpenes carnosic acid (CA) and carnosol, the major oxidation product of CA [21,22]. Research on the identification of new compounds contained in rosemary extracts focuses on their potential antioxidant, anti-inflammatory or anti-cancer activities. However, many other unknown molecules may have hypertrophy-promoting activity in skeletal muscle.

Human myoblasts (i.e., skeletal muscle progenitor cells) can proliferate and at confluence, spontaneously fuse and differentiate, giving rise to myotubes (i.e., quiescent multinucleated cells that express muscle-specific structural proteins). These myotubes can respond to hypertrophic or atrophic stimuli by modulating cell fusion or the proteostasis dynamics, i.e., the balance between protein synthesis (hypertrophy) and protein degradation (atrophy), with or without modification of the fusion index (number of myotube nuclei relative to the total number of nuclei) [9,10,13,23,24]. This cell model is therefore useful for selecting natural molecules with hypertrophic activity on the basis of their efficiency and their toxicity. Here, we found that rosemary leaf extracts induce myotube hypertrophy in human primary skeletal muscle cells. By using a bioassay-guided fractionation approach and human skeletal muscle cells, we identified carnosol as a molecule with myotrophic activity. We then evaluated, by Western blotting, carnosol effects on signaling pathways involved in the control of skeletal muscle hypertrophy and atrophy, and found that carnosol inhibits MuRF1 expression and proteasome activity, independently of its activity on the NRF2/antioxidant pathway.

## 2. Materials and Methods

### 2.1. General Experimental Procedure

Flash column chromatography was performed using a Spot Liquid Chromatography Flash Instrument (Armen Instrument, Saint-Avé, France) equipped with a quaternary pump, an UV/visible spectrophotometer and a fraction collector. The ^1^H-NMR, ^13^C-NMR, COSY, HSQC, and HMBC spectra were recorded on a BRUKER Avance III-600 MHz NMR spectrometer. The coupling constants (*J*) were estimated in Hertz.

### 2.2. Reagent and Standards

Cyclohexane (99.8%), dichloromethane (99.9%), chloroform (99.9%), 1-butanol (99.7%) (HPLC grade), deuterated chloroform (99.8%) and DMSO (99.9%) were purchased from Sigma-Aldrich (Steinheim, Germany). Acetonitrile (99.9%) was purchased from Chromasolv (Seelze, Germany). Formic acid (98%), ethyl acetate (99%) and acetone (99.5%) were from Panreac (Barcelona, Spain). Ethanol (99.9%) was purchased from VWR BDH Prolabo (Monroeville, PA, USA). Carnosol (98%) was purchased from Extrasynthese (Genay, France), and CA (99.5%) from PhytoLab GmbH & Co. KG (Vestenbergsgreuth, Germany).

### 2.3. Plant Material

Wild *Rosmarinus officinalis* plants were collected in the North of Montpellier (France) in February 2015. Dry leaves were ground and directly extracted. Voucher specimens were deposited as no RL05515O.

### 2.4. Extraction

A total of 150 g of ground rosemary leaves were macerated in the dark at room temperature with 900 g of absolute ethanol and 450 g of distilled water, with manual agitation every 24 h. After 7 days, the leaves extract was filtered. Evaporation under reduced pressure to dryness yielded 69 g of hydroethanolic extract, named RLE. The raw dry extract was kept at −20 °C until analysis and purification.

### 2.5. Bioassay-Guided Isolation of Carnosol from the Rosemary Leaf Extract

At each purification step, fractions were tested using the assay described below (part 2.9). The RLE (69 g) was solubilized in water (400 mL), then partitioned into cyclohexane (C_6_H_12_), dichloromethane (CH_2_Cl_2_), ethyl acetate (EtOAc), and butanol (BuOH) (2 × 200 mL for each solvent). Among the five fractions, only RLE-1 (soluble in C_6_H_12_, 2.27 g) could induce myotube hypertrophy and was then fractionated by normal-phase flash column chromatography (Merck Chimie SVF D26-SI60, 15–40 μm-30 g, flow rate 10 mL/min, 10 mL/fraction). Elution was completed with mixtures of C_6_H_12_/EtOAc (100:0 to 0:100), and then CH_2_Cl_2_/methanol (100:0 to 70:30 in 1% then 5% stepwise). After thin-layer chromatography analysis, the fractions eluted with cyclohexane/EtOAc 60:40 to 0:100 (fractions 252–459) were combined and concentrated under reduced pressure, yielding fraction RLE-1-4 (780 mg). RLE-1-4 was purified on LH-20 Sephadex gel (2.4 × 38 cm, 40 g LH-20, elution: 100% CH_2_Cl_2_ to 100% methanol, then 100% acetone, 5 mL/fraction). Fractions 116–141, eluted with CH_2_Cl_2_/MeOH 95:5 to 50:50, were combined and concentrated under reduced pressure, yielding fraction RLE-1-4-5 (225 mg). RLE-1-4-5 was fractionated on normal-phase flash column chromatography (Merck Chimie SVF D26-SI60, 15–40 μm-30 g, flow rate 5 mL/min, 5 mL/fraction) eluted with a mixture of C_6_H_12_/EtOAc (100:0 to 0:100) and then CHCl_3_/methanol (100:0 to 70:30) to obtain 166 fractions. Fractions 29 to 54, eluted with 100% C_6_H_12_, were combined (RLE-1-4-5-2; 90 mg), and deposited again on normal-phase flash column chromatography (Interchim PuriFlash SIHP 15 µm 25 g, flow rate 10 mL/min, 10 mL/fraction, elution C_6_H_12_/EtOAc 100:0 to 0:100). Fractions 38–52 (50 mg) were combined and purified on Sephadex LH-20 gel (1.2 × 24 cm, 20 g, elution CH_2_Cl_2_/methanol 100:0 to 0:100) to obtain 50 fractions. Fractions 17–33 (RLE-1-4-5-2-4-3, 28 mg) were combined and evaporated to dryness. Fraction RLE-1-4-5-2-4-3 was purified by precipitation of impurities with methanol. The soluble fraction in CH_2_Cl_2_ (14 mg) was then analyzed by NMR spectroscopy and identified as pure carnosol: ^1^H NMR (CDCl_3_, 600 MHz), δ_H_ 0.86 (3H, s, 19-CH_3_); 0.91 (3H, s, 18-CH_3_); 1.23 (3H, d, *J* = 2.3 Hz, 17-CH_3_); 1.24 (3H, d, *J* = 2.3 Hz, 16-CH_3_); 1.29 (1H, td, *J* = 3.2–13.5 Hz, 3-CH); 1.68 (1H, m, 3-CH); 1.72 (1H, d, *J* = 5.6, 5-CH), 1.74 (1H, d, *J* = 5.6, 2-CH), 1.88 (1H, t, *J* = 12.9 Hz, 2-CH); 2.02 (1H, dd, *J* = 3.2–13.8 Hz, 6-CH); 2.21 (1H, dt, *J* = 5.0–13.8 Hz, 6-CH); 2.39 (1H, td, *J* = 4.1–13.2 Hz, 1-CH); 2.92 (1H, d, *J* = 13.2 Hz, 1-CH); 3.08 (1H, sept, *J* = 6.7 Hz, 15-CH); 5.37 (1H, d, *J* = 4.1 Hz, 7-CH); 6.64 (1H, s, 14-CH) ^13^C NMR (CDCl_3_, 150 MHz) δ_C_ 19.0 (C2); 19.9 (C19); 22.6 (C16); 22.6 (C17); 27.5 (C15); 29.4 (C1); 29.9 (C6); 31.8 (C18); 34.7 (C4); 41.2 (C3); 45.6 (C6); 48.5 (C10); 78.0 (C7); 112.5 (C14); 121.8 (C9); 132.3 (C13); 132.9 (C8); 141.2 (C11); 141.9 (C12); 175.9 (C20).

### 2.6. High-Performance Liquid Chromatography (HPLC) Analysis

Chromatographic separation and detection for quantitative analysis were performed on an Ultimate 3000 apparatus that included a quaternary pump, a degasser, an automatic sampler, and a DAD detector (Thermo Fisher Scientific Inc., San Jose, CA, USA). The system was operated using the Chromeleon software, version 2.0. Chromatographic separation was achieved on an ODS Hypersyl C18 column (250 mm × 4.6 mm, 5 μm, Thermo Fisher Scientific Inc., San Jose, USA), with a column temperature maintained at 35 °C. Samples (5 mg/mL in methanol) were eluted at a flow rate of 1 mL/min, using solvent A (water/formic acid 99.9:0.1 *v*/*v*) and solvent B (acetonitrile), with a previously described gradient (Morel Saint 2019).

### 2.7. Primary Muscle Cell Cultures

Quadriceps muscle biopsies were obtained from three healthy adults, one 19-year-old male (M19), one 30-year-old male (M30), and one 70-year-old female (F70) at the Centre Hospitalier Universitaire Lapeyronie (Montpellier, France). Donors signed an informed written consent after the description of the protocol (Authorization N° DC-2008-594). Myoblasts (muscle progenitor cells) were purified from the muscle biopsies, and were cultured on collagen-coated dishes in DMEM/F12 medium with 10% fetal bovine serum (FBS), 0.1% Ultroser G and 1 ng/mL of human basic fibroblast growth factor (proliferation medium), as previously described [23]. Myoblasts can differentiate by fusing into long multinucleated myotubes that express specific skeletal muscle proteins. For cell differentiation, confluent cells were cultured in DMEM/F12 medium with 5% FBS (differentiation medium) for 5 days.

### 2.8. Immunofluorescence Staining

Human myotubes were fixed in 2% paraformaldehyde (Electron Microscopy Sciences, Ayguesvives, France) in PBS and permeabilized with PBS/0.25% Triton X-100. Cells were then incubated with a mouse monoclonal anti-troponin T antibody (1/200; Sigma-Aldrich), followed by the Alexa 555-conjugated anti-mouse antibody (1/1000; Thermo Fisher Scientific). Nuclei were revealed by DAPI staining. Images (seven fields for each condition) were taken with a Zeiss epifluorescence microscope, with 5× plan air objective and analyzed with the Image J software.

Myotube area and fusion index analyses:

DAPI-positive cells were analyzed using the Fiji software to obtain the number of nuclei per image, normalized to the total image area to compare all conditions. For this automatic counting, a manual threshold was set to differentiate nuclei from the background, and the number of nuclei was provided by the “Analyze > Analyze particles” command. Troponin-T-positive cells were analyzed using the Fiji software to obtain the total myotube area per image, by setting a threshold to discriminate troponin-T-positive cells from the background and to quantify them with the “Analyze > Measure” command (normalized to the total image area). The fusion index was calculated as the ratio of the number of nuclei in troponin-T positive cells with ≥3 nuclei versus the total number of nuclei. This ratio was calculated by setting a threshold selection, inverting the selection, and pasting it on the DAPI images to highlight only the nuclei of myotubes that were then counted with the same technique used to determine the total number of nuclei.

### 2.9. Western Blotting

Protein extracts (10 or 20 μg/well) were separated by SDS-PAGE gel electrophoresis (25 mA for 1 h) using 12% and 4–15% Mini-PROTEAN Precast Gels, and then transferred (1.3 A, 25 V for 5 min) using the Trans-Blot Turbo Transfer System to nitrocellulose membranes (Trans-Blot Turbo 0.2 μm Nitrocellulose Transfer Pack) (Bio-Rad, Schiltigheim, France). Membranes were blocked at room temperature with Odyssey blocking buffer (EurobioScientific, Les Ulis, France) and probed with the rabbit polyclonal anti-HO-1 (1/2000; ref: 43966S), rabbit monoclonal anti-phosphorylated AKT (serine 473) (1/5000; ref: 4060), rabbit polyclonal anti-AKT (1/5000; ref: 9272), rabbit polyclonal anti-mTOR (1/1000; ref: 2972), rabbit polyclonal anti-phosphorylated mTOR (serine 2448)(1/1000; ref: 2971) (from Cell Signaling, Danvers, MA, USA); mouse monoclonal anti-myosin (skeletal slow) (1/5000; ref: M8421), mouse monoclonal anti-myosin (skeletal fast) (1/5000; ref: M4276), mouse monoclonal anti-troponin T (1/5000; ref: T6277); mouse monoclonal anti-actin (α-sarcomeric) (1/2000; ref: A2172) (from Sigma Aldrich Saint-Louis, MO, USA); mouse monoclonal anti-MAFbx (1/500; ref: SC-166806), mouse monoclonal anti-MuRF1 (1/2000; ref: SC-398608), mouse monoclonal anti-MuRF2 (1/4000; ref: SC-517149) (from Santa Cruz Biotechnology, Dallas, TX 75220, USA); and mouse monoclonal anti-myosin light chain 3 (F310, 1/300; ref: F310-S) antibody from Developmental Studies Hybridoma Bank (Iowa City, IA, USA). Primary antibodies were followed by incubation with IRDye^®^680RD and IRDye^®^800RD secondary antibodies (Eurobio Scientific, Les Ulis, France). Fluorescence was quantified with the Odyssey software. Data were normalized to the Revert bands (Revert Total Protein Stain Kit, LI-COR USA).

### 2.10. RT-qPCR

Total RNA was isolated from cultured human muscle cells using the NucleoSpin RNA II Kit (Macherey–Nagel, Hoerdt, France). The RNA concentration of each sample was measured with an Eppendorf BioPhotometer. cDNA was prepared using the VersocDNA Synthesis Kit (Thermo Scientific, Ilkirch, France). The expression of various genes was then analyzed as previously described [25] by quantitative polymerase chain reaction (qPCR) with a LightCycler apparatus (Roche Diagnostics, Meylan, France), using the following primer sets:
RPPO human forTCATCCAGCAGGTGTTCGRPPO human revAGCAAGTGGGAAGGTGTAATRIM63 human forAGGGACAAAAGACTGAACTGAATATRIM63 human revTCCAGGATGGCATACAACG

### 2.11. Transient siRNA Transfections

MURF1 (siMurf1) and negative control (siCTRL) Silencer Select pre-designed siRNAs were purchased from Fisher Scientific (France). Human myoblasts at confluence were transfected with Lipofectamine RNAiMax according to the manufacturer’s recommendations (Fisher scientific, France), induced to differentiate, and analyzed 72–96 h after transfection.

### 2.12. Proteasome Activity

Myotubes were collected in PBS by centrifugation after scraping from culture dishes. Cell pellets were lysed in 0.5 mL lysis buffer (50 mM TRIS (pH 8), 0.5 mM EDTA, 5 mM MgCl_2_, 0.5% NP40, 2 mM ATP, 1 mM DTT and 10% glycerol) for 30 min, on ice, by vortexing every 10 min. Then, lysates were centrifuged at 15,000 rpm at 4 °C for 15 min. Measurement of proteasome activity: 10 μg of myotube extracts/well for each condition were deposited in 96-well plates and incubated with proteasome substrate alone (Suc-LLVY-AMC- Bachem AG ref-I-1395.0025) or proteasome substrate and the 20S proteasome inhibitor bortezomib (MedChemExp ref: HY-10227/CS-1039) at 20 μM final. The fluorescence was read using a 380/460 nm filter set with the TECAN infinite 200Pro plate reader after 15, 30 and 60 min of incubation. The mean fluorescence for each extract and each condition was calculated, as well as the residual fluorescence (fluorescence of the substrate—(fluorescence of the substrate + the inhibitor)).

### 2.13. Statistical Analysis

Statistical analyses were done with the GraphPad Prism 6.0 software (GraphPad Software Inc., San Diego, CA, USA). Error bars represent the standard deviation (SD) of the mean. Statistical significance was determined using one-way ANOVA; * *p* < 0.05, ** *p* < 0.01, *** *p* < 0.001, and **** *p* < 0.0001 were considered significant.

## 3. Results

### 3.1. Rosemary Leaf Extract Has a Strong Muscle Hypertrophic Activity

We used human skeletal muscle cell cultures to determine whether a rosemary leaf extract (RLE) could induce muscle hypertrophy. Briefly, at day 3 of differentiation, when myoblasts have fused and myotubes have formed, the RLE was added at 15 μg/mL. After 2 days, cells were fixed and the expression of troponin T, a cytoskeletal protein expressed exclusively by myotubes, was analyzed by immunofluorescence. Then, the area of troponin T-positive myotubes was determined with the Fiji software. Compared with controls (vehicle alone), incubation of myotubes with the RLE led to the appearance of large, branched myotubes, resulting in increased muscle cell area, as shown by troponin T immunostaining (Figure 1A,B), and unchanged fusion index (Figure 1C).

### 3.2. Bioassay-Guided Isolation and Activity of the Myotrophic Compound from the RLE

We then used this bioassay to guide the purification and identification of the compound(s) responsible for the RLE myotrophic activity. At each purification step, we evaluated the ability of the obtained fractions to induce myotube hypertrophy. RLE (69 g) was partitioned between water and cyclohexane (C_6_H_12_), dichloromethane (CH_2_Cl_2_), ethyl acetate (EtOAc), and butanol (BuOH) (Figure 2). We found that the C_6_H_12_ soluble fraction (RLE-1, 2.27 g) induced myotube hypertrophy. Then, we purified the fraction RLE-1 by alternating different columns and chromatographic systems (Sephadex LH-20 gel and normal-phase flash chromatography). Six steps of purification were necessary to isolate 14 mg of the active compound (Figure 2). Analysis of the ^1^H-, ^13^C- NMR, 2D-NMR spectroscopy data and comparison with the literature allowed identifying this compound as carnosol [26,27].

Fractions highlighted in gray displayed hypertrophic activity in primary muscle cell cultures.

Incubation of human myotubes with carnosol increased their size without significant induction of cell fusion (Figure 3A–C) starting from 0.5 µg/mL (1.5 μM) (Figure 3D). We then compared the activity of CA and of its oxidized derivative carnosol. Despite the very similar structure of these two diterpenes, only carnosol displayed a myotrophic effect (Figure 3E). To characterize carnosol myotrophic effect, we analyzed by Western blotting the expression level of several skeletal muscle proteins (Figure 3F). We found that carnosol increased the level of slow myosin heavy chain, but not of fast myosin heavy chain, troponin T, sarcomeric actin, and myosin light chain F3. This indicated that carnosol stimulates muscle hypertrophy by targeting the expression of specific proteins.

### 3.3. NRF2 Activation by Carnosol Is Not Sufficient to Induce Myotube Hypertrophy

As carnosol can activate the antioxidant NRF2 pathway in several cell types (not tested in skeletal muscle cells), we determined whether carnosol could induce the NRF2 signaling pathway in skeletal muscle by monitoring the expression of heme oxygenase-1 (HO-1), one of its targets. Quantification of the Western blotting data showed that HO-1 protein level was increased in myotubes incubated with carnosol, compared with untreated control cells, but not as much as with dimethyl fumarate (DMF), a well-known activator of the NRF2 pathway [28] (Figure 4A). However, the myotube area was increased only by incubation with carnosol and not with DMF (Figure 4B), indicating that DMF did not induce myotube hypertrophy. Then, we inactivated NRF2 by using two anti-*NRF2* siRNAs (siNrf2-91 and siNrf2-93), and evaluated HO-1 expression by Western blotting. In cells in which *NRF2* was silenced, HO-1 induction by carnosol was significantly reduced compared with siCTRL myotubes (Figure 4C). However, the two *NRF2* siRNAs did not prevent the carnosol-mediated increase in myotube size (a sign of hypertrophy) (Figure 4D). These results demonstrate that activation of the NRF2 signaling pathway by carnosol is not sufficient to mediate its effect on myotube size.

### 3.4. Carnosol Inhibits Protein Degradation by Targeting the Ubiquitin Proteasome System

To test whether carnosol induced muscle hypertrophy by stimulating protein synthesis or by inhibiting protein degradation, we first investigated its effect on the Akt/mTORC1 pathway that regulates protein synthesis. When Akt is activated, it phosphorylates the mTORC1 complex, ultimately, increasing the translation of target mRNA transcripts. Here, we did not detect any change in Akt phosphorylation level upon incubation of myotubes with carnosol compared with control and also after co-incubation with insulin, a well-known inducer of Akt phosphorylation (Appendix A). Similarly, mTOR phosphorylation level was not affected by incubation with carnosol (Appendix A). These results strongly suggested that the Akt/mTOR pathway is not activated in myotubes in response to carnosol exposure.

We then analyzed the ubiquitin-proteasome system (UPS) that controls protein degradation. Quantification the protein levels of the muscle-specific E3 ligases MuRF1, MuRF2 and MAFbx after incubation with carnosol showed a significant reduction of MuRF1, but not of MuRF2 and MAFbx expression (Figure 5A). Reverse transcription-quantitative PCR analysis indicated that *MuRF1* mRNA levels decreased in a dose-dependent manner after incubation with increasing concentrations of carnosol (Figure 5B). As the RLE increased myotube size, we determined whether it could inhibit MuRF1. Thus, we tested MuRF1 protein level after incubation with the RLE (2 or 15 µg/mL), carnosol (1 μg/mL) or CA (1 µg/mL), which unlike carnosol, has no effect on myotube size. After 2 days, quantification of the Western blotting data showed that 15 µg/mL of RLE was necessary to decrease MuRF1 protein level. Carnosol, but not CA, also significantly decreased MuRF1 protein level. To determine the importance of MuRF1 inhibition in carnosol myotrophic effect, we inactivated MuRF1 by siRNA (siMuRF1). The siMuRF1 specifically and efficiently inhibited MuRF1, but not MuRF2 expression (Figure 5D). Moreover, *MuRF1* silencing significantly increased the myotube area (Figure 5E), indicating that *MuRF1* silencing is sufficient to induce muscle hypertrophy in skeletal muscle cells.

Inhibition by carnosol of MuRF1, an UPS component, strongly suggested a decrease of proteasome activity in myotubes incubated with carnosol. To test this hypothesis, we directly measured in vitro the chymotrypsin-like activity of the proteasome in myotubes incubated with carnosol or transfected with siMuRF1. The proteasome activity was significantly decreased in both conditions (Figure 6A,B).

To confirm the UPS importance in skeletal muscle hypertrophy, we tested the effect of bortezomib, a proteosome activity inhibitor. Incubation with increasing concentrations of bortezomib led to significantly larger myotubes, indicating that inhibition of the proteasome can induce human myotube hypertrophy (Appendix A).

These results strongly suggest that carnosol induces skeletal muscle hypertrophy by repressing the UPS-dependent protein degradation pathway through inhibition of the E3 ubiquitin ligase MuRF1.

## 4. Discussion

Here, we found that RLE was sufficient to induce hypertrophy of human muscle cells and that carnosol, a phenolic diterpene, was the active component mediating this effect in young and aged skeletal muscle cells. Recently, it has been reported that carnosol has anti-cachectic effects mainly by decreasing muscle and adipose tissue loss, in vitro in C2C12 mouse myoblasts incubated with conditioned medium from C26 colon carcinoma cells or with pro-inflammatory cytokines, and in vivo in C26 tumor cell-bearing mice [29]. Moreover, carnosol can regulate glucose homeostasis in skeletal muscle via AMPK-dependent GLUT4 glucose transporter translocation [30], and could be potentially used against insulin resistance and type 2 diabetes mellitus. All these findings indicate that carnosol is a promising molecule for the treatment of skeletal muscle disorders such as aging and cachexia.

Carnosol has various therapeutic effects (anti-cancer, anti-inflammatory and antioxidant activities) [31]. Some of carnosol antioxidant effects are mediated through activation of the NRF2 signaling pathway [32]. NRF2 is a transcription factor that mediates the intracellular antioxidant response by binding to the antioxidant response element in the promoter of its target genes, thus inducing the expression of a set of antioxidant enzymes, called ‘phase 2 enzymes,’ including HO-1 [33]. Regulation of NRF2 signaling preserves the redox homeostasis and protects the skeletal muscle structure and function, thus making of NRF2 an interesting potential target of carnosol effects in muscle cells [34]. We found here that carnosol activates the NRF2 signaling pathway in human primary muscle cells, but that this is not required for its hypertrophic effect. This suggests that carnosol exhibits distinct antioxidant and hypertrophic activities.

To elucidate how carnosol induces hypertrophy in human primary muscle cells, we first looked at the PI3K/Akt/mTOR cascade that represents the major pathway of muscle hypertrophy. Our data clearly indicated that carnosol did not affect Akt or mTOR phosphorylation/activation levels, in agreement with the study by Vlavcheski et al. showing that in L6 rat skeletal muscle cells, carnosol regulates glucose homeostasis, but does not affect the PI3K-Akt pathway [30]. Conversely, in C2C12 myoblasts incubated with conditioned medium from C26 colon carcinoma cells or pro-inflammatory cytokines, and in C26 tumor cell-bearing mice, carnosol activates the Akt signaling pathway and ameliorates cachexia-induced muscle atrophy [29]. These data suggest that carnosol activates the Akt signaling cascade in cells exposed to pro-inflammatory factors, such as in cachexia, but not in normal muscle cells.

Our data suggest that the hypertrophic effect of carnosol in human muscle cells is mediated through inhibition of the E3 ubiquitin ligase MuRF1 and of the proteasome. In muscles, proteins can be degraded through at least four pathways: lysosomes, calcium-activated proteases, ATP-dependent, and ATP-independent proteolytic pathways. The ATP-dependent pathways include the UPS, the most important proteolytic pathway to mediate muscle atrophy [35]. Ubiquitination, which is required for protein targeting to the 26S proteasome complex, involves the coordinated action of the E1 ubiquitin-activating enzyme, E2 ubiquitin-conjugating enzymes, and E3 ubiquitin ligases. In muscles, the two E3 ubiquitin ligases MAFbx and MuRF1 are considered the main markers of muscle atrophy. MuRF1 belongs to a small family with three members in mammals (MuRF1/TRIM63, MuRF2/TRIM55, and MuRF3/TRIM54). MuRF1 is expressed at low levels in adult muscles in normal conditions, and is upregulated by catabolic stimuli. Studies in knock-out (KO) mice revealed that MuRF1 and MAFbx are excellent markers of muscle atrophy, but only MuRF1 is important to preserve muscle mass in pathological contexts [36]. However, MuRF1 does not seem to be required for the basal skeletal muscle metabolism. Like MuRF1-KO mice, MuRF2-KO mice do not have an obvious phenotype in standard conditions. However, most MuRF1/MuRF2 double KO mice die prematurely, and the survivors develop cardiac and skeletal muscle hypertrophy [37].

Several interacting partners have been identified for MuRF-1, mainly myofibrillar/Contractile proteins such as myosin heavy chain (MHC), titin, myosin binding prot-C (MyBP-C), the light chain of myosin (MLC), alpha-actin (a-actin) and telethonin [36,38]. We found that carnosol increased the level of slow myosin heavy chain, but not of fast myosin heavy chain, troponin T, sarcomeric actin, myosin light chain F3 (in this manuscript) and telethonin (data not shown). This indicated that carnosol stimulates muscle hypertrophy by targeting the expression of specific proteins, especially MHCs, a identified interacting partner of MuRF1. Characterisation of myosin heavy chain (MHC) type is a useful method to classify fiber types based on the relationship between MHC type and fiber function. In humans, type I, or slow-twitch, fibers possess slower twitch speeds and are relatively fatigue resistant. Type II, or high-twitch, fibers, present higher twitch speeds than type I fibers but are less fatigue resistant [39]. Thus, carnosol by promoting type I muscle fibers, should make the skeletal muscle fatigue resistant and more enduring. This hypothesis is currently being evaluated on animal models fed with a diet enriched with carnosol.

Here, *MuRF1* silencing in human muscle cells led to an increase of the myotube size. This phenotype was due to *MuRF1* silencing because the siRNA did not affect MuRF2 expression. Therefore, it appears that in a cell system that mimics muscle regeneration in vitro, MuRF1 can control the myotube size. It has been hypothesized that MuRF1 upregulation leads to muscle atrophy by increasing the proteasome activity and protein degradation. As carnosol represses both MuRF1 mRNA and protein expression and induces myotube hypertrophy, we examined the relationship between MuRF1 expression and proteasome activity in human muscle cells. We found that carnosol or MuRF1 silencing by siRNA reduced proteasome activity by 20%. However, modulation of MuRF1 expression does not always coincide with variation in proteasome activity. In several models of atrophy (denervation, hindlimb suspension, functional overload by synergist ablation surgery), there is a disconnection between MuRF1 expression and proteasome activity [1]. We found that in human muscle cells, inhibition of the proteasome by bortezomib (5 nM) is sufficient to induce muscle hypertrophy. This suggests a simple relationship between UPS activity and myotube size during in vitro muscle regeneration. However, many questions remain concerning the relationship between the UPS and muscle differentiation. Proteasome dysfunction impairs satellite cell ability to proliferate, survive, and differentiate, resulting in defective muscle regeneration [40]. Moreover, proteasome component inhibition or knockdown can block myoblast fusion and consequently muscle differentiation [41,42]. Therefore, proteasome activity is not always associated with muscle cell atrophy or hypertrophy, but could be required also for remodeling and growth, depending on the skeletal muscle differentiation stage [35].

How carnosol controls MuRF1 expression remains an open question. Several transcription factors are involved in MuRF1 expression regulation by directly binding to specific sequences in its promoter. FOXO1,3,4 are transcription factors that bind to and induce *MuRF1* promoter activity in several catabolic situations. FOXO factors are downstream of the IGF1/insulin/Akt pathway. In anabolic conditions, Akt phosphorylates and inhibits FOXO proteins, thus repressing MuRF1 expression. However, as we demonstrated that carnosol does not activate Akt, FOXO involvement in MuRF1 repression by carnosol is unlikely. The NF-kB signaling pathway mediates the effects of inflammatory cytokines on muscle wasting. In response to inflammatory cytokines, the IκB kinase complex (IKK) phosphorylates IκB, resulting in its ubiquitination and proteasomal degradation, and then to the nuclear translocation of NF-κB. MuRF1 promoter has NF-κB binding sites [38]. Carnosol inhibits the NF-κB signaling pathway in vitro in C2C12 mouse muscle cells incubated with conditioned medium from C26 tumor cells or pro-inflammatory cytokines. However, this inhibition was significant only at very high concentrations of carnosol (25 μM), whereas in our study 3 μM of carnosol were sufficient to observe an effect on MuRF1 expression [29]. In addition, without incubation with pro-inflammatory cytokine, the activity of NF-κB is very low. Therefore, it seems unlikely that carnosol inhibits MuRF1 by repressing NF-κB in normal muscle cells. Besides FOXO and NF-κB, several transcription factors activate MuRF1: glucocorticoid receptor, p53 protein family, TFEB, myogenin, ZEB1 [1,38]. More work is needed to test whether one of these proteins could be a carnosol target.

Several strategies have been developed to identify natural molecules that could be used for the treatment or prevention of skeletal muscle atrophy by stimulating protein synthesis and/or inhibiting protein degradation. Here, we used a bioassay-guided fractionation approach to identify carnosol as anabolic compound in an RLE. Another strategy targeting molecules that inhibit the gene signatures induced by muscle atrophy allowed the identification of three natural compounds: ursolic acid (a ubiquitous triterpenoid in the plant kingdom, medicinal herbs, also enriched in the skin of apples), tomatidine (a steroidal alkaloid extracted from tomatoes), and apigenin (a trihydroxyflavone abundant in various edible plants including parsley, celery, chamomile, oranges, and grapefruits) [9,10,11]. Many dietary supplements containing ecdysteroids are marketed as “natural anabolic agents”. Extracts of some plants (e.g., *Spinacia oleracea, Rhaponticum carthamoides*) and their active ingredient ecdysterone (20-hydroxyecdysone, 20HE) can significantly increase muscle mass and decrease body fat [12]. Finally, researchers have identified natural molecules that can stimulate mitochondrial and muscle function/mass, such as urolithin A (including ellagitannins precursors isolated from pomegranate) [14] and urolithin B [13], respectively. Our study adds a new plant extract (RLE) and a molecule (carnosol) to this list of natural compounds that may be used for the treatment or prevention of skeletal muscle atrophy through their inhibition of the E3 ubiquitin ligase MuRF1 (and proteasome activity) and protein degradation.

## Figures and Tables

**Figure 1 nutrients-13-04190-f001:**
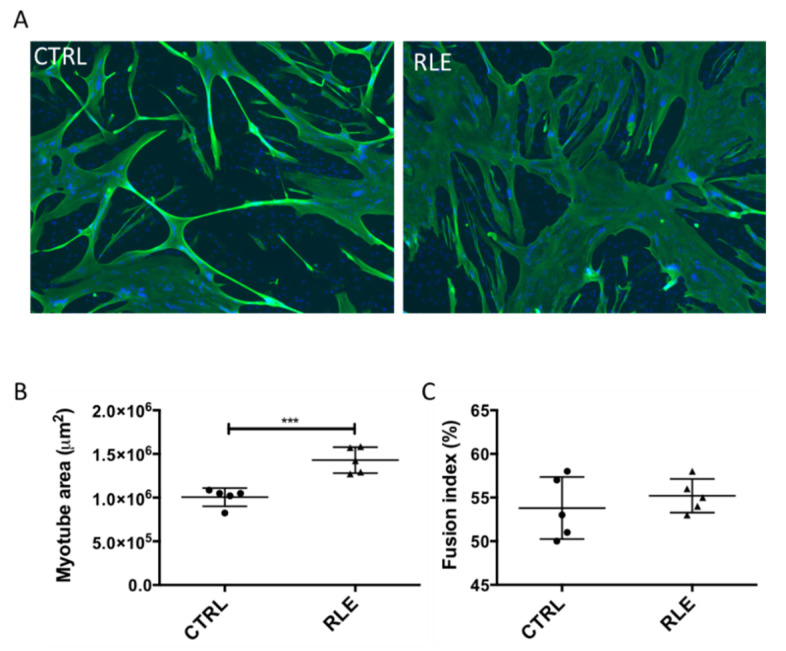
Rosemary leaf extract increases myotube cell size. At confluence, myoblasts (M19) were induced to differentiate for three days, and then, rosemary leaf extract was added (RLE) or not (CTRL) at the concentration of 15 μg/mL. Two days later, myotubes were fixed and analyzed by indirect immunofluorescence with an antibody against troponin T. Nuclei were visualized by DAPI staining (blue). (**A**) Representative images showing troponin T (green) expression. (**B**) Quantification of troponin T staining (myotube area in μm^2^). (**C**) Quantification of the fusion index (percentage). Data are the mean ± SD of five replicates; *** *p* 0.001. M19, 19-year-old male donor.

**Figure 2 nutrients-13-04190-f002:**
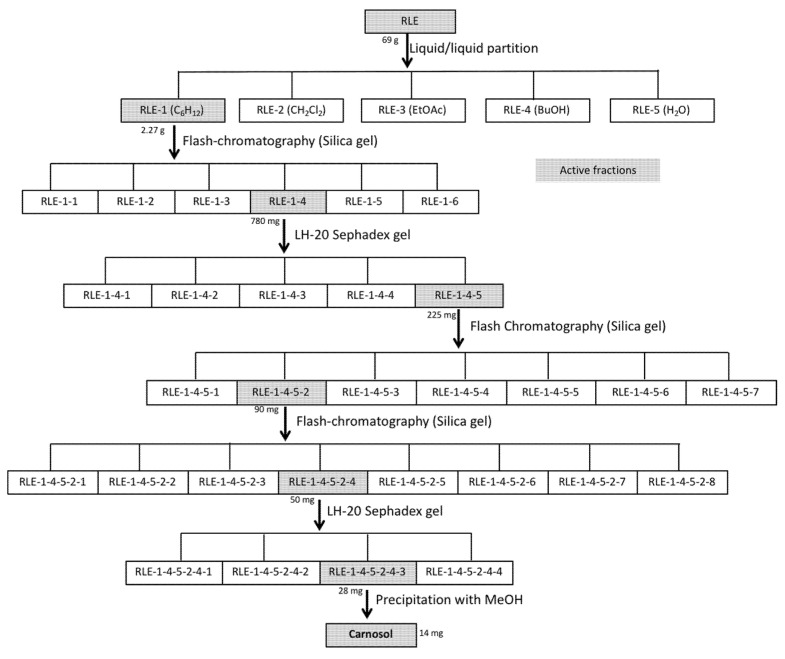
Different steps of carnosol purification from the rosemary leaf extract.

**Figure 3 nutrients-13-04190-f003:**
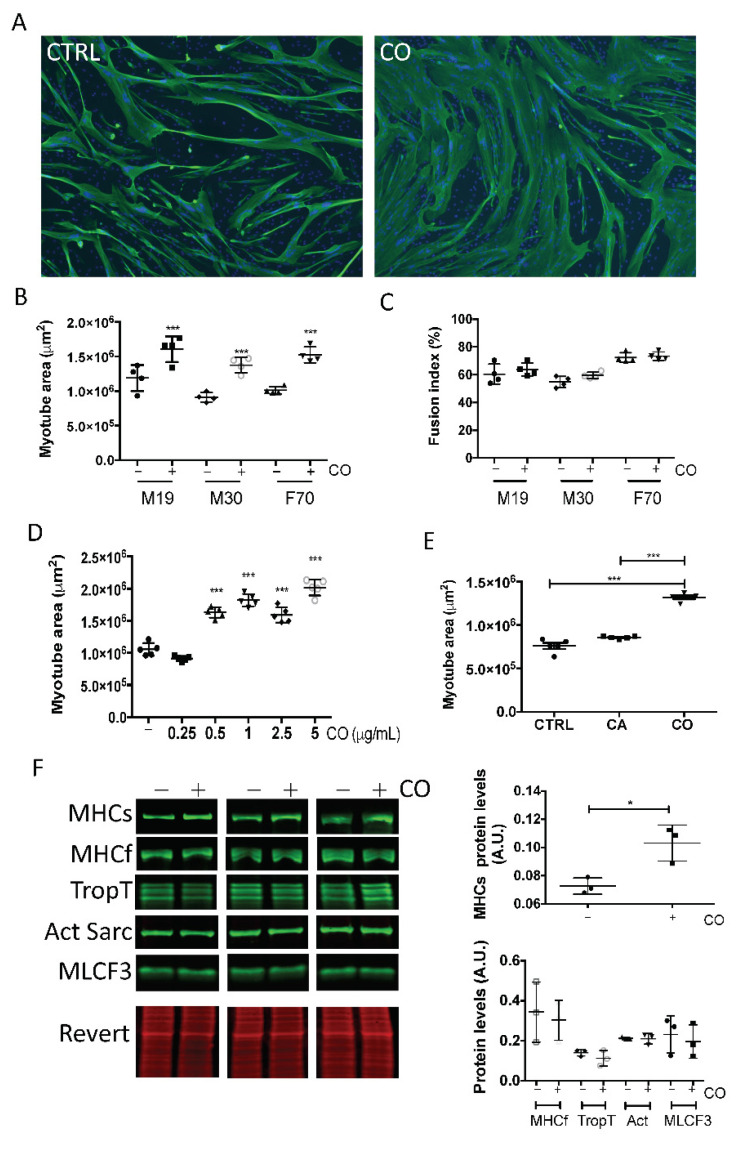
Carnosol induces myotube hypertrophy. At confluence, myoblasts from different donors [M19 in (**A**–**E**) and M19, M30 and F70 in (**B**,**C**)] were induced to differentiate for three days, and then, carnosol was added (CO) or not (CTRL) to the culture at a concentration of 1 μg/mL (**A**–**C**,**E**,**F**), or at increasing concentrations, from 0.25 to 5 μg/mL (**D**). Two days later, myotubes were processed for (**A**) indirect immunofluorescence analysis with an antibody against troponin T (green). Nuclei were visualized by DAPI staining (blue). Representative images of M19 cells. (**B**,**D**,**E**) Quantification of troponin T images (myotube area in μm^2^). (**C**) Quantification of the fusion index (percentage). (**F**) Left: After cell incubation (+) or not (−) with 1 μg/mL carnosol, myotube protein extracts were analyzed by Western blotting with anti-slow myosin heavy chain (MHCs), -fast myosin heavy chain (MHCf), -troponin T (TropT), -sarcomeric actin (Act Sarc), and -myosin light chain F3 (MLCF3) antibodies. Revert^®^ total protein stain was used as internal loading control. Right: quantification of the Western blotting data. A.U., arbitrary units (i.e., the ratio of the level of the protein of interest to the Revert^®^ stain level. * *p* < 0.05, *** *p* < 0.001 (***). Data are the mean ± SD of three to five replicates. M19, 19-year-old male donor; M30, 30-year-old male donor; F70, 70-year-old female donor.

**Figure 4 nutrients-13-04190-f004:**
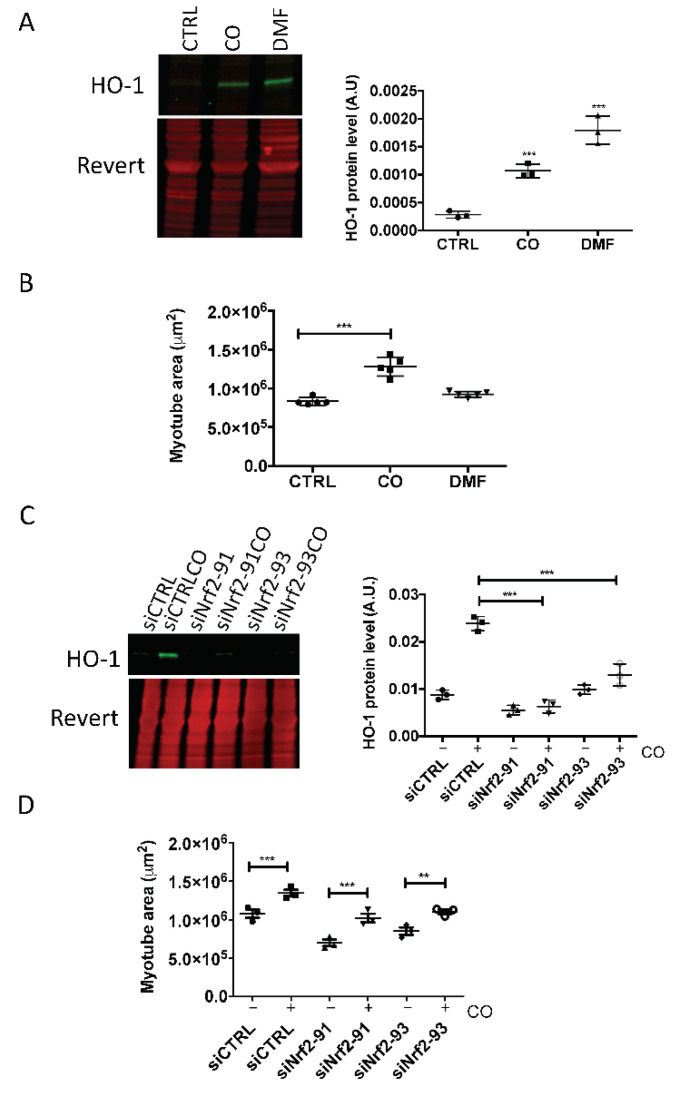
Carnosol induces myotube hypertrophy independently of its ability to activate NRF2. At confluence, myoblasts (M19) were induced to differentiate for 3 days, and then, carnosol (CO, 1 μg/mL, 3 μM) or dimethylfumarate (DMF, 20 μM) was added or not (CTRL) to the culture for 24 h (**A**,**C**) or 48 h (**B**). Then, (**A**) HO-1 protein levels in myotubes were analyzed by Western blotting. Revert^®^ total protein stain was used as internal loading control. On the right, quantification of the Western blotting data; A.U., arbitrary unit (i.e., the ratio of HO-1 protein level to Revert^®^ stain level). (**B**) After incubation with CO or DMF, myotubes were processed for immunofluorescence analysis with anti-troponin T antibodies. Quantification of troponin T immunofluorescence images (myotube area in μm^2^). (**C**) On the left, Western blot analysis of HO-1 levels in myotubes transfected with siNrf2-91 and siNrf2-93 and then incubated (CO) or not with carnosol; on the right, quantification of the Western blotting data; A.U., arbitrary units (i.e., the ratio of HO-1 protein levels to Revert^®^ stain level. (**D**) Quantification of troponin T immunofluorescence images (myotube area in μm^2^) in myotubes transfected with siNrf2-91 or siNrf2-93 and incubated (+) or not (−) with carnosol; ** *p* < 0.01, *** *p* < 0.001. Data are the mean ± SD of three to five replicates. M19, 19-year-old male donor.

**Figure 5 nutrients-13-04190-f005:**
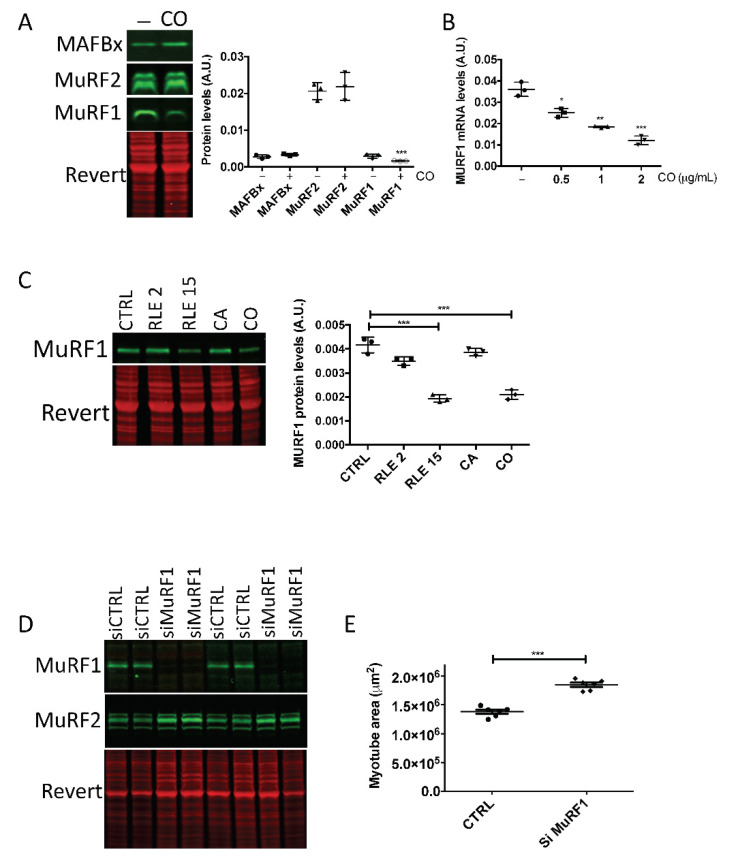
Carnosol inhibits MuRF1 expression. At confluence, myoblasts (M19) were induced to differentiate for 3 days, then, (**A**) carnosol (CO, 1 μg/mL) or (**C**) rosemary leaf extract (RLE, 2 μg/mL or 15 μg/mL), carnosic acid (CA, 1 μg/mL) were added or not (CTRL) to the cultures for 48 h. Myotube protein extracts were then analyzed by Western blotting with (**A**) anti-MAFbx, -MuRF2 and -MuRF1 antibodies and (**C**) anti-MuRF1 antibodies. Revert^®^ total protein stain was used as internal loading control. On the right, quantification of the Western blotting data; A.U., arbitrary units (i.e., the ratio of MAFbx, MuRF2 or MuRF1protein levels to the Revert^®^ stain level). (**B**) RT-qPCR analysis of MuRF1 gene expression in human myotubes incubated or not (CTRL) with 0.5, 1 or 2 μg/mL of carnosol (CO) for 24 h. (**D**,**E**) Myotubes transfected with siMuRF1 and analyzed by (**D**) Western blotting with anti-MuRF1 and -MuRF2 antibodies, and (**E**) by immunofluorescence with an anti-troponin T antibody to quantify the myotube area (in μm^2^); * *p* < 0.05, ** *p* < 0.01, *** *p* < 0.001. Data are the mean ± SD of three to seven replicates. M19, 19-year-old male donor.

**Figure 6 nutrients-13-04190-f006:**
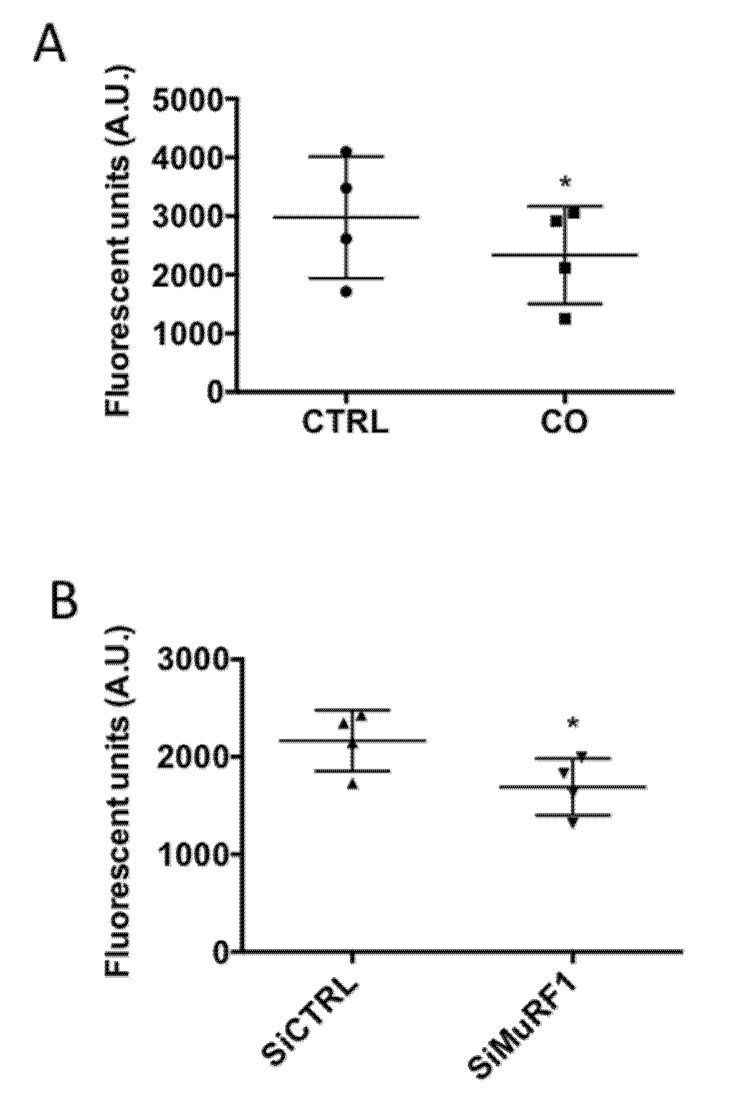
Proteasome activity is inhibited by carnosol. The proteasome chymotrypsin-like activity in protein extracts from myotubes (M19) (**A**) incubated (CO) or not (CTRL) with carnosol (1 μg/mL) for 48 h or (**B**) transfected with a control siRNA (siCTRL) or a siRNA against MuRF1 (siMuRF1) was measured by monitoring the hydrolysis of Suc-LLVY-AMC peptide through quantification of AMC fluorescence in the presence or not of bortezomib, a specific proteasome inhibitor; * *p* < 0.05. Data are the mean ± SD of four replicates. M19, 19-year-old male donor.

## Data Availability

Data is contained within the article and the Appendix A.

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
