# Peer review of "A Bioassay-Guided Fractionation of Rosemary Leaf Extract Identifies Carnosol as a Major Hypertrophy Inducer in Human Skeletal Muscle Cells"

_nutrients, 2021, doi:10.3390/nu13124190_

Round 1
Reviewer 1 Report
This is a nice study that comprehensively assesses the effects of rosemary leaf extract (RLE) and specifically one of its constituents, carnosol, on inducing hypertrophy in human-derived muscle cells. This is an important study that identifies a possible therapeutic agent that could be used to prevent muscle loss. The authors present a series of well thought out experiments to confirm that carnosol promotes hypertrophy in vitro, and try to understand its mechanism of action. They show that while carnosol appears to activate NRF2, this protein does not mediate carnosol's hypertrophic effect. Additionally they rule out Akt/mTOR as mediators of this effect. The eventually show that carnosol reduces Murf1 levels, which in turn results in increased myofibre size. Finally, given the role of Murf1 as a UPS component, they confirm that both carnosol treatment and Murf1 knockdown inhibit proteasome activity.
I have a few minor corrections, comments, and questions -
- Some of the figures describe experiments using cells derived from just one subject. In these cases I would hesitate to use the term "biological replicates" and just use "replicates"
- In Figure 3F its stated that 3-5 replicates were used. But I only see 3 points for each group in the plots. Additionally, the first plot in 3F shows a significant collective increase in MHC expression induced by carnosol. However, the second plot shows no effect on the individual components. Could the authors address this apparent discrepancy (or correct my possible misunderstanding)?
- In Figure 4D I notice a decrease in mytoube area as a result of NRF2 knockdown compared to siCTRL without carnosol treatment. Could the authors please comment on the cause of this apparent atrophy?
Typo -
- line 68 - change "at the origin" to "the origin"
Author Response
Review 1
Comments and Suggestions for Authors
This is a nice study that comprehensively assesses the effects of rosemary leaf extract (RLE) and specifically one of its constituents, carnosol, on inducing hypertrophy in human-derived muscle cells. This is an important study that identifies a possible therapeutic agent that could be used to prevent muscle loss. The authors present a series of well thought out experiments to confirm that carnosol promotes hypertrophy in vitro, and try to understand its mechanism of action. They show that while carnosol appears to activate NRF2, this protein does not mediate carnosol's hypertrophic effect. Additionally they rule out Akt/mTOR as mediators of this effect. The eventually show that carnosol reduces Murf1 levels, which in turn results in increased myofibre size. Finally, given the role of Murf1 as a UPS component, they confirm that both carnosol treatment and Murf1 knockdown inhibit proteasome activity.
I have a few minor corrections, comments, and questions -
- Some of the figures describe experiments using cells derived from just one subject. In these cases I would hesitate to use the term "biological replicates" and just use "replicates"
We changed “biological replicates” to “replicates”
- In Figure 3F its stated that 3-5 replicates were used. But I only see 3 points for each group in the plots.
We agree with the reviewer, Figure 3F shows only 3 replicates. When we mention in the legend of Figure 3 that “Data are the mean ± SD of three to five replicates”, this encompasses the data for the various panels in Figure 3.
Additionally, the first plot in 3F shows a significant collective increase in MHC expression induced by carnosol. However, the second plot shows no effect on the individual components. Could the authors address this apparent discrepancy (or correct my possible misunderstanding)?
We found that carnosol increased the level of slow myosin heavy chain (first plot), but not of fast myosin heavy chain, troponin T, sarcomeric actin, myosin light chain F3 (second plot) in this manuscript) and telethonin (data not shown). This indicated that carnosol stimulates muscle hypertrophy by targeting the expression of specific proteins, especially slow myosin heavy chain, a previously identified interacting partner of MURF1. Characterisation of Myosin heavy Chain (MHC) type is a useful method to classify fiber types based on the relationship between MHC type and fiber function. In humans, type I, or slow-twitch, fibers possess slower twitch speeds and are relatively fatigue resistant. Type II, or high-twitch, fibers, present higher twitch speeds than type I fibers but are less fatigue resistant. Thus, carnosol by promoting type I muscle fibers, should make the skeletal muscle fatigue resistant and more enduring. This hypothesis is currently being evaluated on animal models fed with a diet enriched with carnosol.
The discussion section lines 513-555 was modified in consequence.
- In Figure 4D I notice a decrease in mytoube area as a result of NRF2 knockdown compared to siCTRL without carnosol treatment. Could the authors please comment on the cause of this apparent atrophy?
Previous studies have found that deficiency of Nrf2 exacerbated muscle loss in old but not young mice. Nrf2 deficiency and age induced mitochondrial dysfunction, increased markers of oxidative stress and excessive autophagy activation in skeletal muscle, which can be a potential mechanism for the development of sarcopenia. We found that inactivation of Nrf2 in human muscle cells induces myotube atrophy. We did not study the molecular causes of this atrophy, but we can speculate that the accumulations of oxidative damage accompanied by enhanced autophagy are responsible for this atrophy induced by the inactivation of Nrf2.
Typo -
- line 68 - change "at the origin" to "the origin"
It seems to us that “at” is correctly placed, but I will let the editor check this particular point.
We would like to thank the reviewer for its constructive criticisms and suggestions, and we are confident that we have now addressed to the majority of its concerns.
Reviewer 2 Report
Dear authors,
I would like to congratulate you on completing such an interesting project. Muscle mass loss has been lately become a main focus in aging and clinical population which requires various remedies in order to tackle this issue. Overall, this manuscript is well written; however, I have some minor comments which would help to improve the quality of the manuscript.
Introduction
Line 41- "Therefore, muscle loss prevention may contribute to improve the quality of life and to attenuate chronic diseases and mortality. "
i would recommend replacing "prevention" to reduction/limitation as it may not be possible to fully prevent it.
Line 46- "However, compliance with exercise programs remains low and their implementation is difficult"
Please elaborate as to how implementation of exercise programs are difficult. Is it really difficult?
Considering the big picture of muscle hypertrophy and all factors involved, how much of impact rosemary can have i terms of dosage, duration of consumption and etc...?
What are the main practical applications of this research project? This should be clearly stated.
Author Response
Review 2
Dear authors,
I would like to congratulate you on completing such an interesting project. Muscle mass loss has been lately become a main focus in aging and clinical population which requires various remedies in order to tackle this issue. Overall, this manuscript is well written; however, I have some minor comments which would help to improve the quality of the manuscript.
Introduction
Line 41- "Therefore, muscle loss prevention may contribute to improve the quality of life and to attenuate chronic diseases and mortality. "
I would recommend replacing "prevention" to reduction/limitation as it may not be possible to fully prevent it.
We changed to reduction.
Line 46- "However, compliance with exercise programs remains low and their implementation is difficult"
Please elaborate as to how implementation of exercise programs are difficult. Is it really difficult?
To take into account reviewer comments, we modified the sentence “However, compliance with exercise programs remains low and their implementation is difficult” to “. However, compliance with exercise programs and their implementation are difficult. These are partly the consequence of patient pathologies (fatigue, pain) which limits their participation in long-term physical activity compared to people without chronic illness”. We added a new reference.
Considering the big picture of muscle hypertrophy and all factors involved, how much of impact rosemary can have i terms of dosage, duration of consumption and etc...?
It is very difficult to answer this question fully. Nevertheless, we can try to emit a certain number of hypotheses. Carnosol is found in considerable quantities in Rosemary such as approximately 0.2–1% in dried Rosemary and until 10% in commercially available Rosemary extracts. According to the work of Lu et al, 2021, injection by IP of 10 mg/Kg/day of carnosol for 6 days (equivalent to 500 mg/Kg/Day of rosemary leaf extract if 0.2% carnosol) is sufficient to attenuate muscle atrophy by cancer cachexia in mice. We are doing experiments on mice lasting 3 months to evaluate the effectiveness of food enriched with carnosol on skeletal muscle function.
What are the main practical applications of this research project? This should be clearly stated.
We added line 473: “All these findings indicate that carnosol is a promising molecule for the treatment of skeletal muscle disorders such as aging and cachexia.”
We would like to thank the reviewer for its constructive criticisms and suggestions, and we are confident that we have now addressed to the majority of its concerns.